# Basic and Preclinical Research for Personalized Medicine

**DOI:** 10.3390/jpm11050354

**Published:** 2021-04-29

**Authors:** Wanda Lattanzi, Cristian Ripoli, Viviana Greco, Marta Barba, Federica Iavarone, Angelo Minucci, Andrea Urbani, Claudio Grassi, Ornella Parolini

**Affiliations:** 1Fondazione Policlinico Universitario A. Gemelli IRCCS, 00168 Rome, Italy; wanda.lattanzi@unicatt.it (W.L.); cristian.ripoli@unicatt.it (C.R.); viviana.greco@unicatt.it (V.G.); marta.barba@unicatt.it (M.B.); federica.iavarone@unicatt.it (F.I.); angelo.minucci@policlinicogemelli.it (A.M.); andrea.urbani@unicatt.it (A.U.); claudio.grassi@unicatt.it (C.G.); 2Dipartimento Scienze della Vita e Sanità Pubblica, Università Cattolica del Sacro Cuore, 00168 Rome, Italy; 3Dipartimento di Neuroscienze, Università Cattolica del Sacro Cuore, 00168 Rome, Italy; 4Dipartimento di Scienze Biotecnologiche di Base, Cliniche Intensivologiche e Perioperatorie, Università Cattolica del Sacro Cuore, 00168 Rome, Italy

**Keywords:** personalized medicine, stem cells, mesenchymal stem cells, induced pluripotent stem cells, neuroscience, proteomics, genomics, proteogenomics

## Abstract

Basic and preclinical research founded the progress of personalized medicine by providing a prodigious amount of integrated profiling data and by enabling the development of biomedical applications to be implemented in patient-centered care and cures. If the rapid development of genomics research boosted the birth of personalized medicine, further development in omics technologies has more recently improved our understanding of the functional genome and its relevance in profiling patients’ phenotypes and disorders. Concurrently, the rapid biotechnological advancement in diverse research areas enabled uncovering disease mechanisms and prompted the design of innovative biological treatments tailored to individual patient genotypes and phenotypes. Research in stem cells enabled clarifying their role in tissue degeneration and disease pathogenesis while providing novel tools toward the development of personalized regenerative medicine strategies. Meanwhile, the evolving field of integrated omics technologies ensured translating structural genomics information into actionable knowledge to trace detailed patients’ molecular signatures. Finally, neuroscience research provided invaluable models to identify preclinical stages of brain diseases. This review aims at discussing relevant milestones in the scientific progress of basic and preclinical research areas that have considerably contributed to the personalized medicine revolution by bridging the bench-to-bed gap, focusing on stem cells, omics technologies, and neuroscience fields as paradigms.

## 1. Introduction

Basic research, by definition, advances fundamental milestones in scientific knowledge, generating new ideas, posing new hypotheses, and validating them in experimental models. In the current stage of scientific development, basic research has been contributing to found the development of novel biomedical applications. In particular, the huge and rapid development of genomics research in recent decades has boosted the development of personalized medicine.

The DNA sequence constituting the human genome differs by as little as 0.1% across individuals [1]. This small difference in the primary structure, along with diverse epigenetic modifications and multifactorial molecular interactions, is responsible for the wide phenotypic dissimilarity uniquely characterizing each human being that includes physical traits, disease susceptibility as well as responsiveness to drugs [2]. Therefore, the traditional approach of “one-drug-fits-all” to treat human diseases is nowadays outdated, prompting the emergence of personalized strategies sustained by research efforts at multiple levels.

With the aim to customize medical care and cures based on individual characteristics, personalized medicine has been progressing in recent years to embrace a wider range of biomedical fields and cross-domain expertise, bridging the bench-to-bed gap toward the fulfillment of integrated patient profiling, disease modeling, and targeted drug design. In this scenario, the progress achieved in basic and preclinical research is constantly providing novel tools and valuable pieces of scientific knowledge [3]. On the one hand, personalized medicine can nowadays rely on multi-parametric and multi-modal profiling of patients through high-throughput and highly scalable omics technologies. On the other hand, it requires preclinical experimentally tractable models recapitulating as much of the human pathophysiology, representing an unevaluable opportunity to identify preclinical stages of diseases, to build diagnostic and prognostic practices, make adequate differential diagnoses and exploit innovative biotechnologies in the design of advanced targeted therapies [3].

This paper aims to review the scientific progress in specific basic and preclinical research areas that have considerably contributed to the most recent personalized medicine revolution from different standpoints. The following three main topics will be addressed: (i) the advances in stem cells’ research to study disease mechanisms and design advanced therapies, (ii) the evolving field of integrated omics technologies for translating structural genomics information into actionable knowledge, and (iii) the scientific milestones achieved in neuroscience research as a paradigm of preclinical research applied in personalized medicine (see Figure 1).

Being not directly related to one another, the selected broad areas represent non-overlapping paradigms of the inherent heterogeneity and cross-domain nature of personalized medicine research and are all enabling turning basic research results into applications. Particularly, these research areas are among the valuable scientific efforts spent in basic and translational research at the Fondazione Policlinico Universitario A. Gemelli IRCCS in Rome (FPG). The research focus and mission of our institution is indeed devoted to the evolving field of personalized medicine along with the advanced biotechnologies subsidiary to its development. Toward the fulfillment of these objectives, FPG indeed relies on integrated cross-domain expertise, covering basic, translational, and clinical science areas, implemented in the development of novel diagnostic and therapeutic strategies to advance the progress of personalized medicine and bridge the bench-to-bed gap.

## 2. Stem Cells in Personalized Medicine

Stem cells are key tools to fulfill an improved understanding of disease biology and the design of innovative biological treatments tailored to individual patient genotypes/phenotypes. Indeed, the rapidly growing amount of scientific data on stem cell biology is uncovering their role and involvement in the pathogenesis of diverse conditions characterized by tissue degeneration and/or decreased endogenous healing capabilities. On the other hand, stem cells and their derivatives are being implemented in innovative therapeutic strategies to restore damaged tissues and organs.

By definition, stem cells self-renew their own pool through symmetric cell division while maintaining the potentiality to differentiate into morpho-functionally specialized cell lineages upon receiving inductive signals. Within an adult organism, heterogeneous pools of multi-, oligo-, and uni-potent stem and progenitor cells are found to support tissue homeostasis in physiologic conditions and regeneration after injury throughout life. Adult stem cells act in synergy with their regulatory-specific microenvironment, defining the stem cell niche as the functional unit of tissue regeneration [4]. Stem cells resting quiescent inside the niche can be activated in response to tissue injury to promote regeneration by producing proliferating units while initiating lineage commitment through asymmetric cell divisions [5]. The array of signals involved in stem cell regulation widely depends on individual factors that vary with patients’ genotype, aging, exercise, disease status, along with additional complex multifactorial environmental cues. In this section, we will discuss the most relevant aspects and applications of stem cells under the observational lens of personalized medicine.

### 2.1. Stem Cells in Regenerative Medicine Applications

Regenerative medicine is a multidisciplinary field that aims at developing strategies to regenerate, repair, improve or replace cells, organs, and tissues that have been damaged by diverse *noxae* (e.g., trauma, inflammatory and immune-based diseases, surgery, congenital defects, aging). The increased knowledge on location and functions of tissue-specific niches guided stem cell research toward the development of novel targeted strategies to heal structures and restore biological functions following tissue injury, and delay disease progression [6]. Niche homeostasis can be indeed disrupted, owing either to direct damage to the quiescent stem cell reservoir or to damage of tissue microenvironment integrity and signaling, leading to impaired self-healing capabilities. The individual response in maintaining stem cell niche integrity and functionality may underlie significant differences in clinical manifestations and therapeutic response/outcomes. Table 1 summarizes the state-of-the-art on main stem cells’ applications in regenerative medicine, further discussed in this paragraph.

The best-characterized adult stem cells translated in clinical applications are the multipotent hematopoietic stem cells (HSCs) and mesenchymal stromal cells (MSCs), along with the oligopotent epidermal stem cells (EPSCs), briefly discussed below.

HSC transplantation represents the current treatment for a number of hematopoietic disorders, such as malignancies (to replenish bone marrow after ablation by chemotherapy or radiotherapy), hemoglobinopathies, and immunodeficiency syndromes, including primary monogenic disorders [7,8,9]. Interestingly, the substantial progress in HSC engineering research and manufacturing has allowed the market approval of genetically engineered HSC as a personalized treatment for the severe combined immunodeficiency due to adenosine deaminase deficiency (ADA-SCID) [10], a representative example of patient-personalized regenerative medicine application based on HSC. In particular, for ADA-SCID patients without matched donors, the gene therapy is based on the stable transduction of HSC through lentiviral or retroviral vectors. This treatment led to a survival rate of 100%, surpassing the efficacy of the allogeneic HSC transplantation with fully matched donors [10].

MSCs are extensively tested as therapeutic strategies in a much broader range of applications for tissue regeneration [11,12,13,14,15] and for inflammatory conditions [16,17,18,19,20,21]. MSCs exploit their regenerative effect by increasing their own pool and providing renewed niche components by both differentiating into mesodermal progenitors and attracting supporting cells [5,22]. Although their plasticity driving tissue regeneration is a key property, MSCs’ therapeutic effect is mainly a result of their potent immunomodulatory functions, which make them potentially suitable also in allogeneic transplantation [23,24]. MSC exert most of their properties by secreting paracrine factors with pleiotropic functions [25,26,27] both as free soluble bioactive factors and as the content of extracellular vesicles (EV), mainly exosomes, whose composition varies depending on the tissue environment with related external signals [5,13,28,29,30,31,32]. This MSC secretome has considerable advantages over cell-based strategies for manufacturing, safety, handling, storage, shelf life, hence potential translation into cell-free biological therapies [33]. Marked immune-modulatory properties are particularly a hallmark of perinatal-derived MSCs, including those from the umbilical cord, decidua [34,35,36,37,38], chorionic villi [39,40], and fetal membranes [18,23,41,42,43,44]. In this regard, human amniotic membrane MSCs (hAMSCs) and their secretome have been successfully applied in different preclinical models where inflammation occurs, including lung [45,46,47,48] and liver [49] fibrosis, wound healing [50,51,52], collagen-induced arthritis [53,54], multiple sclerosis, inflammatory bowel disease, sepsis [53], colitis [53,55], traumatic brain injury [56], and Huntington’s disease [57].

Currently, there are over 20 clinical trials (excluding those with unknown status) evaluating placenta-derived cells and placenta MSC registered on the NIH Clinical Trials website (https://clinicaltrials.gov/, accessed on 10 March 2021). The published or current clinical trials are either Phase I, II, or III and include a variety of inflammatory disorders, such as pulmonary idiopathic fibrosis [58], peripheral artery disease, Crohn’s disease [59,60], multiple sclerosis [61], diabetes [62], ischemic stroke, pulmonary sarcoidosis [63], active rheumatoid arthritis, and muscle injury due to hip arthroplasty [64].

Strategies have been tested, in recent years, to optimize MSC therapeutic efficacy by induced genetic modifications with the goal to enhance their potential and tailor MSC-based therapy to patients’ specific needs. Several attempts have been made to genetically engineer MSCs from different sources to modify their phenotype and expression profiles (e.g., the expression of cytokines, signaling molecules, and growth factors) hence improving their properties, using viral and non-viral gene transfer techniques (reviewed by Damasceno and colleagues [65]). Most strategies were aimed at tuning MSC immune modulation properties depending on the target clinical end-point [66]. Another use of MSC as a personalized medicine device is to exploit them as “Trojan horses” to defeat cancers by delivering anticancer drugs to affected sites [67,68]. In this regard, the tumor tropism can be increased in MSCs by gene transfer to induce the over-expression of homing molecules (such as the chemokine receptor 4, CXCR4), which proved successful in mice [69]. Novel nano-biotechnologies and biomaterials are being tested as MSC-priming stimuli, able to enhance the viability and biological activity of transplanted cells, and by promoting paracrine secretion, also acting on the composition and concentration of regulatory miRNAs in the secreted exosomes [70].

Finally, EPSCs are long-lived stem cells that can extensively self-renew and replenish terminally differentiated keratinocytes during skin physiological modeling and in wound healing. Based on their powerful regenerative potential, EPSCs have been studied since 1970, then introduced over 30 years ago in the clinical practice to implement cultured epidermal autografts for the treatment of chronic skin wounds and extensive burns [71]. The transplantation of gene-corrected EPSCs for the treatment of epidermolysis bullosa represents a valuable paradigm of personalized regenerative medicine, originally introduced with the first successful clinical trial in 2006 [72]. Extended follow-up studies allowed demonstrating that the transgenic EPSCs were stably integrated into the epidermal layers for years and enabled regenerating the entire skin in treated patients [73,74].

### 2.2. Stem Cells as Tools for Understanding Diseases

Somatic stem cells have been widely exploited to study disease pathophysiological mechanisms at a basic level. Nonetheless, some critical aspects, such as the origin and inherent heterogeneity of tissues, the different sensitivity to culture conditions, or the variation in donor characteristics, limited the efficacy of using cellular models. These concerns have been partially overcome with the advent, over a decade ago, of induced pluripotent stem cell (iPSC) technology, which allows reprogramming patient-specific somatic cells (e.g., skin fibroblasts and blood cells) toward a pluripotent state by introducing the “Yamanaka factors” (named after Shinya Yamanaka who first engineered these cells in his lab), namely, the four transcription factors Oct3/4, Sox2, Klf4, and c-Myc. These are usually introduced by overexpressing them using retroviral vectors [75,76]. iPSC introduction was worth the Nobel prize in 2012 since it boosted the development of personalized medicine by providing patient-specific cell-based autologous therapies [77]. Currently, the foremost utility of human iPSCs (hiPSCs) is in the development of optimized disease models as platforms for drug discovery and testing, ranging from 2D cultures, through 3D models, to organoids and human–rodent *chimeras*. To this aim, iPSCs derived from patients (e.g., with monogenic disorders) are induced ex vivo to effectively differentiate cell phenotypes that faithfully mimic and recapitulate hallmarks of the disease [77,78]. In particular, iPSC technology offers the unprecedented opportunity to model diseases associated with specific cell types/tissues that are hardly obtained from living patients, such as neurons, cardiac and pancreatic cells, among others. The iPSC technology has indeed found application above all in the in vitro modeling of neurological diseases, as discussed in Section 4.3.

The implementation of iPSCs in personalized medicine has dramatically benefited from the recent advancements in DNA editing techniques, i.e., the 2020 Nobel-awarded development of the clustered regularly interspaced short palindromic repeats (CRISPR)/CRISPR-associated system 9 (Cas9) genome engineering tool [79]. CRISPR/Cas9-editing applied to the iPSC genome is widely exploited in translational precision medicine research for the development of novel cell-based therapeutic strategies and in the early phases of drug discovery [80,81,82,83,84]. Precisely, the artificial CRISPR system for gene editing consists of two essential components, a guide RNA (gRNA) to target complementary DNA sequences for cleavage and the CRISPR-associated protein (Cas), an RNA-guided DNA endonuclease found in Streptococcus pyogenes, which generates site-specific double-strand breaks based on the gRNA-defined sequence [85,86]. For instance, CRISPR/Cas9 technology is introduced in iPSC production to engineer ex vivo somatic cells isolated from patients diagnosed with Mendelian disease to correct the genetic defect, hence obtaining a pool of pluripotent cells with a healthy genotype to serve for advanced cell therapy applications. As detailed in Section 4.1, genome editing is widely exploited in neuroscience research to study otherwise inaccessible human brain tissues and model brain disorders.

Also, genome editing enables the introduction of gene mutations into wild-type cells to study the pathogenicity of selected gene variants, assess their effect on cellular differentiation, tissue development, and morphogenesis, and design targeted molecular therapies [87,88,89,90,91]. Recently, a hiPSCs-based drug, bosutinib, has been approved by the Food and Drug Administration (FDA) for the treatment of chronic myelogenous leukemia [92].

Finally, the combination of human iPSCs with gene editing to develop 3D organoids has recently provided a more powerful cellular resource for the design of regenerative medicine strategies and model systems. Organoids are 3D structures in a dish, usually assembled as spherical cell culture masses including different cell lineages, to mimic the tissue organization, cellular compartmentalization, and organ-like functions [79]. Their development was stimulated by the need for more accurate model systems to study integrated processes occurring in the complex human tissue environment to overcome the limitations deriving from species-specificity of diseases and developmental mechanisms while reducing animal sacrifices. The implementation of genetically corrected human iPSCs in the assembly of organoids allow generating transgenic tissues as sources for organ replacement therapies [93]. Brain, retina, inner ear, stomach, intestine, thyroid, lung, liver, and kidney organoids have been produced to date [94]. Currently, stem cell-derived healthy and diseased organoids are exploited for the development of personalized therapies and to study patient-specific drug response [95], enabling disease modeling in a more physiological environment by combining the structural and functional complexity of a patient organ. Organoid-based patient-customized models have been developed for complex disorders, such as neurological diseases (i.e., microcephaly, autism), inflammatory diseases (i.e., ulcerative colitis and Crohn’s disease) [96,97,98,99], and monogenic disorders, such as cystic fibrosis [100,101], Huntington’s disease and polycystic kidney disease [102,103].

## 3. OMICS Technologies in Personalized Medicine

OMICS represents the key technological advances that have led the development of personalized medicine by providing an unprecedented amount of data enabling to dissect the molecular basis of many diseases and tracing detailed patients’ molecular signatures on a system biology scale.

Since its emergence around 2010, next-generation sequencing (NGS) has introduced a breakthrough in clinical genomics laboratories providing fundamental structural information on potential actionable molecular elements. However, with its massive sequencing capacity, NGS enables identifying an enormous number of structural genomic variants, whereas the proportion of functionally classified variants has not been increasing at the same rate. Therefore, an ever-increasing interpretive gap broadens the challenge on variant interpretation [104].

A well-established example is the deep sequencing of the breast cancer-associated (BRCA) 1 and 2 genes. Germline mutations in the *BRCA* genes predispose individuals to develop breast and ovarian cancer (OC), as well as increasing the risk of developing other types, such as prostate and pancreas cancers [105,106,107]. Germline pathogenic variants (PVs) of *BRCA* genes have been identified in about 15% of women with OC and about 5% of all breast cancers, respectively [108,109,110].

Functional studies have shown that BRCA1/2 proteins act in response to cell stress through activation of DNA repair processes [111]. The BRCA pathway is also involved in Fanconi anemia, a rare hereditary disease caused by genetic defects in DNA repair proteins and characterized by genomic instability with increased cancer risk [112].

*BRCA* gene testing is widely exploited in clinical practice as a paradigm of personalized medicine, especially in the context of OC [113,114]. BRCA mutated patients are administered prophylactic treatments, such as risk-reducing agents, bilateral mastectomy, and salpingo-oophorectomy, to prevent cancer recurrence. Targeted therapeutics, such as poly (ADP-ribose) polymerase inhibitors (PARPi), are also used for these patients and prove effective in selected BRCA mutated cases [115,116].

The interpretation of the functional effects of *BRCA* variants remains a significant limitation in *BRCA* testing. The American College of Medical Genetics and Genomics and the Association for Molecular Pathology standards and guidelines provided a detailed framework for the interpretation of genomic variants [117], though the classification of variants of unknown significance (VUS) remains a relevant challenge. The Evidence-based Network for the Interpretation of Germline Mutant Alleles (ENIGMA) consortium has joined classification efforts by diverse international laboratories, clinical cancer geneticists, and variant database curators to address this challenge [118]. VUS can be missense substitutions, small in-frame deletions or insertions, or silent coding and intronic alterations that may influence splicing or gene expression regulation, with unknown functional effects, which impedes their classification as either “pathogenic variant (PV)” or “not PV”.

The classification of a BRCA VUS is based on multiple approaches: higher frequency in cases than in controls, phylogenetic conservation of the modified amino acid, Grantham score for missense variants, co-segregation of the VUS with the disease or with a PV in the same family, abnormal transcripts and functional assays. In the absence of sufficient information, assessment of pathogenicity relies on in silico analyses and assays that measure effects on gene or protein functions. In silico analyses assume that changes in evolutionarily conserved amino acids are most likely to be pathogenic, but prediction algorithms are less effective than well-validated functional assays [119]. Despite considerable efforts to determine their pathogenicity, VUS interpretation remains a conundrum in most cases, causing *BRCA* testing not to support clinicians in establishing a personalized treatment and increasing anxiety in patients and their families due to etiological uncertainty.

*BRCA* genetic testing in OC patients is a paradigm for genotype-driven personalized therapeutics. All reported BRCA pathogenic variants in OC patients lead to homologous recombination deficiency (HRD; i.e., mutations that impair DNA damage repair mechanisms causing loss or gain of chromosomal regions). HRD tumors show increased sensitivity to PARPi agents; the majority of OC patients could be treated with a PARPi, in both treatment and maintenance settings. Interestingly, there are currently several functional assays that allow accurate assessment of the effects of VUS on gene function [120], which can be hence exploited to select PARPi-sensitive OC and establish personalized care strategies. In addition, the most common acquired mechanism of resistance to PARPi appears to be through the restoration of homologous recombination [121].

Given the polygenic risk associated with cancer etiopathogenesis and relying on the high throughput of NGS technologies, the guidelines established by the American Society of Clinical Oncology stated that *BRCA* testing should be complemented by the targeted sequencing of additional multigene panels, including at least the *RAD51C*, *RAD51D*, *BRIP1*, *MLH1*, *MSH2*, *MSH6*, *PMS2,* and *PALB2* genes [122]. In fact, some of these genes are also linked to the risk of developing other types of cancer. For example, the Fanconi anemia gene known as partner and localizer of BRCA2 (*PALB2*) is associated with a significant risk of breast cancer [123], whereas PVs in the mismatch repair genes (*MLH1, MSH2, MSH6,* and *PMS2*), known to cause Lynch syndrome, lead to microsatellite instability, which in turns increases the risk of ovarian, endometrial and colon cancers [124]. Mutations in these genes are also useful in pharmacogenomics, as they may influence the susceptibility of cancer to chemotherapeutics, such as platinum [125,126].

On the other hand, cancers are characterized both by cytological diversity, being composed of dynamic populations of different cell types constantly and rapidly evolving, and consequently by functional diversity [127]. Intra-tumoral heterogeneity is a common feature of several types of cancer, resulting from the interaction between genetic heterogeneity and the tumor microenvironment that stimulates the activation of different transcriptional programs [128,129]. This heterogeneity represents a challenge for the development of targeted drug treatments [130]. In this regard, recent advances in NGS allowed the possibility of sequencing the entire genome and transcriptome of individual cells, opening new perspectives toward characterizing the functional heterogeneity of the tumor. In the last few years, a number of high-throughput single-cell RNA-sequencing (scRNA-seq) platforms have been increasingly introduced to enable transcriptomic profiling at a single-cell resolution in order to better unravel the mechanisms of cancer pathogenesis and identifying novel putative therapeutic targets [131].

### 3.1. The Proteome Setting in Functional Genomics Mapping

The perspective of the absence of a dominant actionable variant, as shown in the case of *BRCA* testing in OC, provides a clear understanding of the urgent need for OMICS analysis at a functional level. A deeper understanding of how the structural genomic information can drive the phenotype and of how genome-encoded functions are performed and modulated at the proteome level can make a significant contribution to the development of targeted therapies. In this context, proteogenomics combines next-generation DNA and RNA sequencing with mass spectrometry to provide an in-depth characterization of tumor profiles [132]. In particular, in recent years, many efforts in preclinical research have been devoted to the understanding of the proteome modulation by post-translational modifications (PTMs), such as phosphorylation, mainly for proteins associated with chromosomal instability.

Specifically, the Clinical Proteomic Tumor Analysis Consortium (CPTAC) has extended the boundaries of proteogenomics and phosphoproteomics to the in-depth and comprehensive characterization of HGSC (high-grade serous ovarian carcinoma) tumors to profile the complicated ovarian HGSC phenotype, correlating differences in protein and PTMs levels with clinical and genomic data.

Tumor samples typically show a generalized increase in phosphorylation hence in the activation of related pathways, compared with normal tissues. In particular, comprehensive comparisons, based on a combined proteogenomic and phosphoproteomic approach, have been applied on HGSC and matched normal precursor tissue samples (Fallopian tube, FT) [133,134]. HGSC tissues appeared to be enriched in proteins associated with DNA repair and DNA replication regulation mechanisms (e.g., homologous recombination, inter-strand crosslink repair, regulation of DNA damage response, negative regulation of telomere maintenance, DNA-dependent DNA replication maintenance of fidelity). In contrast, protein networks related to immune functions and developmental processes, such as muscle processes, immune function, and reproductive and neurological signaling pathways, appeared to be consistent in normal FT compared to tumors [133].

With regard to other PTMs, particular emphasis was given to acetylation. Besides representing a typical epigenetic modification, protein acetylation (Ac), specifically lysine acetylation, has been implicated in the regulation of cellular metabolism [135]. It is currently well known that Ac is a modification of several proteins, both histone and non-histone, located in diverse cellular compartments, such as the nucleus, the cytoplasm, and the mitochondrion, and playing various functions ranging from gene regulation, cell signaling to metabolism under physiological and pathological conditions [136].

Along with HRD and histone modifications, the study of proteome-wide acetylation dynamics has been exploited to stratify patients for personalized therapies. Specifically, in this context, the simultaneous acetylation of K12 and K16 on histone H4 resulted in a possible alternative functional biomarker of HRD phenotype [114,134].

Therefore, especially in the absence of a dominant driving mutation, analysis at a functional level, including PTMs, could be extremely useful for a deeper understanding of the biology driving cancers such as HGSC. In this context, high-resolution proteomics represents a powerful tool to unravel the potential role of proliferation-induced replication stress in promoting the characteristic chromosomal instability of HGSC, guiding personalized therapeutic strategies in a functional system framework.

### 3.2. Technological Challenges in Proteogenomics

Proteogenomics represents a novel development of omics technologies that integrate genomic, transcriptomic, and proteomic data to define functional correlations between genes and proteins. However, in this scenario, there is a limitation: proteomic data are not as abundant as genomic data. In the last years, different approaches have been followed to overcome this lack by searching the best platforms to integrate and correlate omics data correctly. Assays such as reverse phase protein arrays (RPPA) have made it possible easily to collect semi-quantitate larger numbers of proteins at a time. This is described as a high-throughput technology for the quantitative measurement of hundreds of proteins in biological and clinical samples [137]. In RPPA, protein lysates spotted onto nitrocellulose coated glass slides are detected by an immune-based assay. This array format enables the quantification of a selected protein or phosphoprotein in multiple samples under the same experimental conditions at the same time. SOMAscan^®^ assay is another detection system developed with the idea of rapidly quantifying a set of proteins. This assay is described as a proteomic tool for discovering biomarkers, for preclinical and clinical drug development, and for clinical diagnostics applied to different diseases and conditions. It is based on affinity capture and uses synthetic DNA SOMAmers (slow off-rate modified aptamers) as protein capture reagents rather than antibodies [138].

Even if both RPPA and SOMAscan have been promoted as the key to the large-scale identification and quantification of proteins, with a putative immediate clinical translation into diagnostics, they offer some limitations. Both techniques rely on antibodies or synthetic SOMAmers whose specificities are not fully characterized and currently do not cover the whole-proteome, including PTMs. For this reason, over the past two decades, mass spectrometry (MS)-based technologies have emerged as methods of choice for the confident and near exhaustive identification and quantification of proteins in a biological sample. They gained prevalence in proteomic research, allowing the collection of analytical quality datasets with consistent quantification. MS-based technologies have significantly contributed to unraveling cellular signaling networks, elucidate the dynamics of protein-protein interactions in different cellular states, and improving diagnosis and molecular understanding of disease mechanisms [139]. MS-based proteomics can reveal the quantitative state of a proteome based on an exact recognition of the primary chemical structure of the proteins or peptides with several PTMs, which are not detected with other methods. They thereby provide insights into the real biochemical condition of the respective cell or tissue. Moreover, the quantification of small molecules through MS technology is consolidated and considered the gold standard for many clinical biochemical applications [140].

However, the limits of discovery proteomics are being overcome by targeted proteomics, exemplified by selected/multiple reaction monitoring (S/MRM) [141] and parallel reaction monitoring (PRM) [142]. These targeting methods provide consistent and accurate quantification, even at low abundances and in complex mixtures. Therefore, targeted proteomics, with its sensitivity and highly quantitative capabilities, is well-suited for hypothesis-driven research and clinical studies where a smaller number of proteins, such as potential biomarkers, are to be measured in large numbers of patient samples [143]. The best combination would be to carry out discovery proteomics using both top-down and bottom-up strategies as the first accurate screening and with the possibility to correctly identify all the different PTMs. Top-down proteomics investigates the intact sequence of the protein under examination, avoiding as much as possible any sample alteration. Bottom-up proteomics is based on sample pre-digestion (typically with trypsin) followed by the analysis of peptide fragments by high-throughput analytical methods [143]. It is necessary to extrapolate complete information on unique proteins that may be potential objectives for targeted quantification. After that, it is possible to transfer the method into targeted proteomics with the aim of being directly applied in the clinical routine, with all the repeatability, robustness, and accuracy features that are needed to be used for diagnostic purposes.

## 4. A Paradigm of Preclinical Research in Personalized Medicine: Neuroscience

A key challenge for international health systems is the growing occurrence of neurodegenerative diseases, primarily associated with the rapid aging of the population due to increased life expectancy. The development of new technologies for large-scale analysis has allowed a deeper understanding of neurodegenerative disorders, encouraging the development of a precision medicine approach for these diseases. Advance in the knowledge of the molecular mechanisms involved in higher brain functions, such as cognition, learning, and memory, goes hand in hand with the development of new technologies. Indeed, one of the major obstacles to understanding the function and dysfunction of the brain is the limited access we have to it when compared to other body organs. A further layer of complexity is represented by the intricate networking of billions of neurons characterized by considerable molecular, morphological, and functional heterogeneity. Last but not least, brain function is markedly affected by the interaction of neurons with glial cells that play a critical role in signal propagation along nerve fibers as well as in the transmission of information within the neural circuits. The complex scenario outlined so far indicates that the altered function of one or more brain components mentioned above may produce a variegated picture of brain dysfunctions also within the same nosographic entity, thus stressing the relevance of personalized medicine in the field of neuroscience. In the last years, this field has greatly progressed thanks to technological advancements discussed in the following sections, and it has widely relied on preclinical studies based on animal models.

Of note, the promise of personalized medicine to cure each patient with a treatment tailored to their disorder first requires a detailed understanding of the mechanisms underlying its pathogenesis.

For a long time, experimental animal models have been representing the only reliable systems to unveil pathophysiological mechanisms of disease, identify biomarkers of its onset/progression and investigate drug efficacy. Models that recapitulate most human neuropathophysiology have been developed, including autism spectrum disorders (ASD) [144], Down’s syndrome (DS) [145], Alzheimer’s disease (AD) (alzforum.org/research-models/alzheimers-disease), Parkinson’s disease (PD) [146], schizophrenia (SZ) [147], and stroke [148], which are just a few representative examples. Research on animal models continues to be essential to understanding brain function and human diseases.

Preclinical neuroscience research is taking great advantage of the discovery of CRISPR-Cas technology as a relatively straightforward, inexpensive, and precise tool to study the molecular and cellular functions of a gene product within an identified cell type in the brain [149]. CRISPR-Cas-based genome editing has been used to manipulate the epigenome as well as to correct mutations in the genome of brain cells [150]. In particular, these genome editing tools have opened the way to model human neurological disorders useful in translational precision neuroscience research. Indeed, CRISPR/Cas9 has been used in human induced pluripotent stem cells (hiPSCs, see below) to introduce: (i) early-onset AD-causing mutations in amyloid precursor protein (APPSwe) and presenilin 1 (PSEN1M146V) [151]; (ii) leucine-rich repeat kinase 2 (LRRK2) G2019S mutation [152], the most prevalent genetic cause of familial and sporadic PD; (iii) mutations in 10 ASD-relevant genes (*AFF2/FMR2, ANOS1, ASTN2, ATRX, CACNA1C, CHD8, DLGAP2, KCNQ2, SCN2A,* and *TENM1*) [153]; and iv) combinatorial perturbation of four SZ-associated risk genes [154].

CRISPR/Cas9 has also been used for generating animal models recapitulating the brain pathology seen in human diseases [155]. Indeed, recently CRISPR/Cas9 has been used to: (i) generate DS models [156]; (ii) reduce levels of SHANK3 protein in macaques, creating genetically engineered non-human-primate models of ASD [157]; and (iii) model LRRK2 G2019S mutation in common marmosets [158].

A classical approach to deliver CRISPR-Cas in vivo takes advantage of the use of viral vectors. Several types of recombinant viruses are now available as a safe and efficient tool for gene delivery [159]. The most commonly used viral vectors in neuroscience are the adeno-associated virus (AAV) from non-pathogenic and non-enveloped replication-defective parvovirus owing to its mild immunogenicity, high infection ability, strong neuronal tropism, and inefficiency to integrate into the human genome [160]. Moreover, specific promoters such as the glial fibrillary acidic protein promoter (GFAP) or the inducible intron human synapsin I promoter (ihSyn1) allow to overexpress or to downregulate a gene of interest selectively in astrocytes and neurons, respectively. AAV-mediated transgene expression was reported throughout the brain >1 year after viral administration [161]. Neuroscientists are now attempting gene therapy using AAV to treat human neurological diseases such as spinal muscular atrophy [162], including two of the most common neurodegenerative diseases such as PD [163] and AD [164] (for a detailed review, see Hudry and collaborators [165]).

### 4.1. Optogenetics and Chemogenetics

Optogenetics and chemogenetics are emerging techniques allowing temporal control of the activity of neurons through an increase or a decrease in neuronal excitability. Optogenetics provides precise temporal control of neuronal activation with light pulses thanks to the transgenic expression of photosensitive ion channels named opsins [166]. In contrast, chemogenetics provides the ability to modulate neuronal activation for several hours with a single administration of small molecules for genetically encoded engineered proteins [167] or a designer drug through the expression of designer receptors activated exclusively by designer drugs (DREADDs) [168]. Specific promoters allow the expression of opsins or chemoactivatable analogs in sub-neuronal populations in selected brain regions [169]. These techniques are useful for testing hypotheses regarding neural circuit mechanisms and evaluating the effect on animal behavior, increasing the knowledge of the physiological role of specific neuronal circuits. Among billions of neurons, this correlation is fundamental to investigate the pathological mechanisms associated with behavioral alterations and consequently to understand diseases with the aim to develop new therapeutic strategies. Chemogenetics is an alternative to optogenetics, especially for applications that require long-term, minimally invasive control and implementation in vivo since efficient illumination of deep brain structures is challenging. There is increasing interest in extending chemogenetics in many applications of neuroscience research and as therapeutics. In particular, innovative systems deriving from synthetic biology are used in various models of human neurological and neuropsychiatric disorders available for preclinical research to overcome the limits of traditional therapeutical approaches [170]. Cell transplantation of bioengineered cells secreting neurotrophic factors has been used in AD patients [171]; however, a synthetic polyglutamine-binding peptide, fused to heat shock cognate protein 70 binding motifs, has been used to tackle mutant Huntingtin protein in Huntingtin disease [172].

New synthetic biology strategies are based on engineered proteins whose activity is spatiotemporally controlled with unnatural and safe cues. The most promising engineering approaches have the advantage of leaving unaffected the overall structure of the proteins, thus conserving their localization and ability to interact with physiological targets [173]. The recent and exciting results obtained by the artificial intelligence algorithm (AlphaFold, developed by DeepMind, deepmind.com (accessed on 2 March 2021)), which determined a 3D protein structure from its primary sequence [174], allow developing engineered analogs of proteins whose crystallographic structure has not been solved yet, thus accelerating the development of innovative biotechnological tools. Both optogenetics and chemogenetics have enormous potential to (i) study the role of specific proteins and intracellular pathways in brain physiology; (ii) identify new molecular and cellular processes that, in nervous circuits, are responsible for higher cognitive functions, with particular reference to learning and memory; and (iii) identify and test new drugs for personalized medicine.

### 4.2. Human Induced Pluripotent Stem Cells (hiPSCs)-Derived Neurons

The unprecedented access to live human neurons from patients via (iPSC) reprogrammed from somatic cells revolutionizes medical research opportunities for neurological and neuropsychiatric disorders. Human neurons obtained from iPSCs expressed mature neurons’ markers, exhibited action potentials, and formed functional synapses [93]. In patients affected by neurological or neuropsychiatric disorders, the same pathology may show variable clinical features, and it is unlikely to benefit from a single treatment. Interestingly, variable pathological phenotypes can stand out using iPSCs-derived neurons because iPSCs retain all of the individual donor’s genetic information and differences in genetic backgrounds [175]. These neurons obtained from chemical and genetic manipulations of peripheral tissue biopsies preserve transcriptomic features of their donors’ age and recapitulate the hallmarks of the disease. Indeed, human neurons obtained from iPSCs of patients affected by neurological disorders, including Alzheimer’s disease and Parkinson’s disease, have been shown to closely recapitulate cellular and molecular features of human diseases [176,177]. Interestingly, the FDA has recently approved clinical trials based on the use of human iPSCs in neurological disorders, namely, Parkinson’s disease [178,179] and spinal cord injury [180].

Other than neurons and glial cells, iPSCs are used to obtain brain organoids that recapitulate in vitro features of early human neurodevelopment, including the generation, proliferation, and differentiation of neural progenitors into neurons and glial cells [181]. It appears clear that an in-depth functional and molecular study of human neurons and glial cells from patients has enormous potential to gain insight into the mechanisms of the disease and to identify and test new drugs for personalized medicine.

Live human neurons could also be used as an advantageous model to identify biomarkers. Recently, exosomes emerged as a novel biological source containing proteins, lipids, and RNAs [182]. Exosomes are microvesicles with a diameter of 30–130 nm, released from most cells. Of note, the exosome cargo is protected from degradation by a lipid bilayer; thus, the interest is increasing for the analysis of the exosome cargo as diagnostic biomarkers of neurological and neuropsychiatric diseases [183]. The growing interest in these vesicles is also due to their presence in biological fluids, which allows their isolation and analysis through a simple blood sample, thus overcoming the limits in accessibility to the brain. The size of exosomes enables them to cross the blood-brain barrier, and fractionation procedures are now available, allowing to isolate the pool of exosomes of neuronal or glial origin from biological fluids [184]. Therefore, studying the content of these vesicles from iPSCs-derived human neurons and patients can provide very important information on cerebral functions and their alterations in neurological and neuropsychiatric diseases.

### 4.3. Patch-Seq

In preclinical neuroscience research, the combination of single-cell reverse transcription PCR with patch-clamp recordings has long been used to enable the correlation of gene expression patterns with the function of brain cells [185,186,187]. However, this combination of techniques is limited to the analysis of some pre-selected genes. The recent advances in whole-transcriptome amplification and the rapid advancement in next-generation sequencing methods combined with patch-clamp recordings have become a promising strategy to identify molecular features associated with specific neuronal functions and phenotypes, overcoming the limitation of an unbiased discovery of novel transcripts and splice variants. The profiling of single-cell transcriptomes by RNA sequencing with an electrophysiological and morphological characterization of individual neurons is known as Patch-seq [187,188,189]. A greater understanding of the correlation among the neuronal morphology and function with gene expression patterns in human neurons from patients is dramatically changing our knowledge in the field and provides increasing opportunities to tailor diagnosis and treatments.

## 5. Conclusions

The growing interest and investment in personalized medicine research are yielding an enormous amount of data and results exploited to achieve a truly holistic view of the individual patient’s phenotype. The specific contributions of basic and preclinical research are translated to help clinicians abating the pitfalls of classical clinical practice, reducing ineffective treatments and complex differential diagnosis flowcharts. Therefore, this new era of integrated medical practice led by in-depth biological knowledge is predicted to enable bringing down healthcare costs.

Firstly, despite the intense genome sequencing effort spent so far, this does not provide a functional insight in specific structural alteration both on a direct relationship and even more on in the perspective of non-linear molecular mechanisms. The genome-to proteome relationships mapping represents a logical evolution of systematic genome sequencing, paving the way to more robust and functional exploitation of omics in personalized medicine. Furthermore, preclinical investigations are enabling the translation of NGS mapping both at the germline and at the somatic level into actionable knowledge in personalized medicine. In this regard, the improved insights into stem cell biology governing tissue homeostasis have made available simplified disease models for first-level preclinical exploration of disease mechanisms. Concurrently, stem cell research paved the way for the development of advanced therapies for patient-tailored regenerative treatments. In addition, the new knowledge and technologies available for the neuroscience research community have opened an unprecedented opportunity to investigate causal links between biochemical signaling and neuronal functions as well as to outline the possibility to use new genetic approaches for personalized therapeutic applications. Of note, the clinical exploitation of the research efforts and outcomes in regenerative medicine and neuroscience in patient-tailored treatments could foster decreasing the health burden related to the aging population.

Taken together, these paramount gains of fundamental knowledge are crucial to reach the promise of more effective therapies, earlier diagnoses, and reduction of adverse events for each patient.

## Figures and Tables

**Figure 1 jpm-11-00354-f001:**
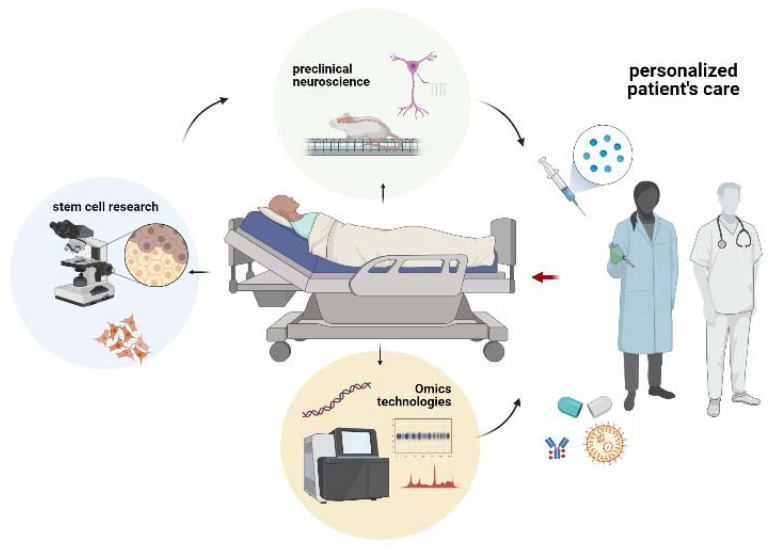
The figure schematizes how personalized medicine starts from basic and preclinical studies. In particular, somatic and stem cells from patients are studied as the source of information to dissect diseases at the basic level and develop cell-based treatments, omics technologies enable deciphering specific functional signatures and biomarkers in patient’s biospecimens, neuroscience research supports new technologies for understanding the conundrum of brain functioning and related disorders. Altogether, these three research areas contribute to the development of personalized cures and advanced therapies, enabling physicians with innovative tools for patient’s centered cures and care protocols. Created with BioRender.com (accessed on 2 March 2021).

**Table 1 jpm-11-00354-t001:** Stem cell applications in regenerative medicine.

Stem Cell Type	Applications
Hematopoietic Stem Cells (HSC)	Hemoglobinopathies
	Aplastic anemia
	Lymphoma
	Aggressive lymphomas
	Multiple myeloma
	Immunodeficiency syndromes
	Sickle cell disease
	Myeloid leukemia
	Acute lymphoblastic leukemia
	Monogenic disorders
	ADA-SCID
Mesenchymal Stromal Cells (MSC)	Tissue regeneration
	Burn wounds
	Wound healing
	Bone defects
	Cartilage defects
	Inflammatory disorders
	Pulmonary idiopatic fibrosis
	Peripheral artery disease
	Chron’s disease
	Multiple sclerosis
	Diabetes
	Ischemic stroke
	Pulmonary sarcoidosis
	Reumatoid arthritis
MSC Secretome	Wound healing
	Inflammatory disorders
	Collagen-induced arthritis
	Lung fibrosis
	Liver fibrosis
	Multiple sclerosis
	Sepsis
	Colitis
	Brain injury
	Huntington’s disease
Epidermal Stem Cells (EPSC)	Skin wounds and burns
	Epidermolysis bullosa
Induced Pluripotent Stem Cells (iPSC)	Monogenic disorders
	Drug discovery and testing
	Neurological disease
	Cardiac disease
	Pancreatic disease
	Organ replacement therapies (Brain, retina, inner ear, stomach, intestine, thyroid, lung, liver, kidney)

## Data Availability

Not applicable.

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
