# Peer review of "Basic and Preclinical Research for Personalized Medicine"

_jpm, 2021, doi:10.3390/jpm11050354_

Round 1
Reviewer 1 Report
It is a well written and fair review of the current standard of personalized medicine and focusses on various stem cell research and therapies, the implementation of -omics in the clinic, and advances in neuroscience. This reviewer did not identify concerns.
Author Response
REVIEWER 1
It is a well written and fair review of the current standard of personalized medicine and focusses on various stem cell research and therapies, the implementation of -omics in the clinic, and advances in neuroscience. This reviewer did not identify concerns.
>> (Authors) We wish to thank the Reviewer for the kind comments and appreciation of our manuscript.

Reviewer 2 Report
This is an interesting manuscript, presenting and reviewing specific contemporary aspects of personalized medicine. The paper is well written , scientifically accurate, and quite informative, overall a useful addition to the field and an enjoyable read.
I have one major comment. From the 3 main chapters of the manuscript, 2 are well justified, those on stem cells, and OMICS technologies. However,the third one (neuroscience research in personalized medicine), somehow pops up out of the blue, without obvious connection with the topic of the manuscript except perhaps the research interest of the authors. Especially it is difficult to accept any justification for including a chapter on animal models in the context of this paper.
The title is not representative of the content of the paper. It is too general, while the content eventually focuses on neuroscience research. A better title reflecting the content should be selected.
Overall, the manuscript, while interesting, reads like two different papers collated in one. The authors have to solve this issue, either by adjusting the content to the title or the title to the content.
Author Response
This is an interesting manuscript, presenting and reviewing specific contemporary aspects of personalized medicine. The paper is well written , scientifically accurate, and quite informative, overall a useful addition to the field and an enjoyable read.
>> (Authors) We wish to thank the Referee for her/his kind words of appreciation of our work.
I have one major comment. From the 3 main chapters of the manuscript, 2 are well justified, those on stem cells, and OMICS technologies. However, the third one (neuroscience research in personalized medicine), somehow pops up out of the blue, without obvious connection with the topic of the manuscript except perhaps the research interest of the authors. Especially it is difficult to accept any justification for including a chapter on animal models in the context of this paper.
>> (Authors) We thank the Reviewer for this insightful observation, and we agree that the chapters sounded not clearly related to one another. In the current revised version we have tried to introduce the neuroscience field as a valuable example for preclinical research in personalized medicine. To this end we have implemented the following changes:
- The title of paragraph 4 has been changed to “A paradigm of preclinical research in personalized medicine: neuroscience”
- The discussion has been implemented to justify the importance of disclosing neuroscience applications in personalized medicine (see lines 475-479 and 491-493).
Moreover, based on the Reviewer’s comment, the sub-paragraph on animal models has been removed, while retaining a brief mention to them in the introductory section to emphasize how their use in preclinical studies is crucial when it comes to studies on complex organs such as the brain. On this regard, we have also implemented the introduction to emphasize that the promise of personalized medicine to cure each patient with a treatment tailored to their disorder first requires a detailed understanding of the mechanisms underlying its pathogenesis (see lines 494-509).
The title is not representative of the content of the paper. It is too general, while the content eventually focuses on neuroscience research. A better title reflecting the content should be selected.
Overall, the manuscript, while interesting, reads like two different papers collated in one. The authors have to solve this issue, either by adjusting the content to the title or the title to the content.
>> (Authors) We agree with the Reviewer, and we adjusted the content to the title, as detailed above.

Reviewer 3 Report
Dear Authors,
I enjoyed reading your manuscript, personalized medicine is a goal that many scientists have in mind when conducting a basic research. The topic is very interesting and worth studying.
However, I have some suggestions than may improve your paper:
- Paragraph 2.1 line 125-126 – could you please write more about genetically engineered HSCs – how they are modified, is it lentiviral modification, and also what is the efficiency of this type of treatment
- Paragraph 2.2 line 189 – could you explained “defined factors”
- It would be nice to have a table that summarizes that part, for example type of stem cells versus applications
- In the paragraph 3 NGS technology is described only in terms of identification of BRCA mutations in ovarian cancer. I think that in this context breast cancer should also be mentioned. Also, not only BRCA mutations are studied, NGS technology allowed to introduced multiple risk gene panels that are used in clinical practice for the identification of patients with cancer predisposing gene variants. What is more, single-cell RNAseq technology opened new possibility to study not only intertumor, but also intratumor heterogeneity in cancer.
- Paragraph 3.1 – line 310-331 – I am a bit confused, as Phosphoproteomics is a term related to protein phosphorylation as one type of PTMs, and you focus on histone acetylation which is more related to Epigenomics. Also, a paragraph 324-326 has no citation.
- Paragraph 4.2 – line 450 – is foreign mean that it was introduced by using lentiviral gene delivery?
Author Response
I enjoyed reading your manuscript, personalized medicine is a goal that many scientists have in mind when conducting a basic research. The topic is very interesting and worth studying.
>> (Authors) We wish to thank the Referee for her/his appreciation of our work and tried to address at our best his/her precious suggestions and concerns, as detailed in the following point-by-point list.
- Paragraph 2.1 line 125-126 – could you please write more about genetically engineered HSCs – how they are modified, is it lentiviral modification, and also what is the efficiency of this type of treatment
>> (Authors) these details have been added accordingly (see lines 132-136 of the revised version).
- Paragraph 2.2 line 189 – could you explained “defined factors”
>> (Authors) we have further specified the “Yamanaka factors” typically exploited in the production of iPSCs (see lines 199-203 of the revised version).
- It would be nice to have a table that summarizes that part, for example type of stem cells versus applications
>> (Authors) As suggested, we have introduced, a “Table 1” (see pages 3-4 on the revised version) that summarizes the state-of-the art on main stem cells’ applications in regenerative medicine, discussed in the corresponding paragraph 2.1.
- In the paragraph 3 NGS technology is described only in terms of identification of BRCA mutations in ovarian cancer. I think that in this context breast cancer should also be mentioned. Also, not only BRCA mutations are studied, NGS technology allowed to introduced multiple risk gene panels that are used in clinical practice for the identification of patients with cancer predisposing gene variants. What is more, single-cell RNAseq technology opened new possibility to study not only intertumor, but also intratumor heterogeneity in cancer.
>> (Authors) Paragraph 3 has been modified, accordingly, to implement these aspects, which we agree are crucial on this context. In particular:
- reference to breast cancer and additional details on the pathophysiology of BRCA-related cancer risk have been introduced on lines 272-280.
- a brief discussion on the genetic heterogeneity and related polygenic cancer risk and pharmacogenomic implications, has been implemented on lines 320-330.
- finally we have introduced a reference to novel technologies implementing single cell RNAseq to study inter- and intra-tumor variability due to the activation of different transcriptional programs (see lines 331-343).
- Paragraph 3.1 – line 310-331 – I am a bit confused, as Phosphoproteomics is a term related to protein phosphorylation as one type of PTMs, and you focus on histone acetylation which is more related to Epigenomics. Also, a paragraph 324-326 has no citation.
>> (Authors) we apologize for the confusion and tried to fix this issue by widely reorganizing the paragraph 3.1. In this revised version, the concept of protein acetylation and its putative link to cancer-related mechanisms has been better clarified (see lines 362-389 in the current version of the manuscript).
- Paragraph 4.2 – line 450 – is foreign mean that it was introduced by using lentiviral gene delivery?
>> (Authors) We agree on the need for improved clarity and replaced “foreign” with “transgenic” (on line 510 of the current revised version).

Round 2
Reviewer 2 Report
My comments are well addressed. I have no further reservations, the manuscript is ready for publication.